# Pharmacologic Induction of BRCAness in *BRCA*-Proficient Cancers: Expanding PARP Inhibitor Use

**DOI:** 10.3390/cancers14112640

**Published:** 2022-05-26

**Authors:** Rachel Abbotts, Anna J. Dellomo, Feyruz V. Rassool

**Affiliations:** 1Department of Radiation Oncology, University of Maryland School of Medicine, Baltimore, MD 21201, USA; anna.dellomo@som.umaryland.edu (A.J.D.); frassool@som.umaryland.edu (F.V.R.); 2University of Maryland Marlene and Stewart Greenebaum Comprehensive Cancer Center, Baltimore, MD 21201, USA

**Keywords:** DNA repair, homologous recombination, PARP inhibitor, synthetic lethality, BRCA mutations, BRCAness, epigenetic therapy, kinase inhibitor, cell cycle inhibitor

## Abstract

**Simple Summary:**

BRCA1 and−2 are critical components of the homologous recombination pathway of DNA repair required to effectively repair DNA double strand breaks, leading to an increased cancer risk in patients with inherited *BRCA* mutations. An additional subset of cancers exhibit ‘BRCAness’, harboring repair defects stemming from mutations in non-*BRCA* DNA repair genes. Both *BRCA*-mutant cancers and cancers with a BRCAness phenotype are sensitive to PARP inhibitors, a class of cancer therapy drugs that inhibit the repair of DNA single strand breaks. To expand the use of PARP inhibitors to a larger group of patients, studies have focused on new combination strategies using agents that can induce BRCAness. This review focuses on the current status of drug-induced BRCAness in combination with PARP inhibitors to enhance cancer treatment.

**Abstract:**

The poly(ADP-ribose) polymerase (PARP) family of proteins has been implicated in numerous cellular processes, including DNA repair, translation, transcription, telomere maintenance, and chromatin remodeling. Best characterized is PARP1, which plays a central role in the repair of single strand DNA damage, thus prompting the development of small molecule PARP inhibitors (PARPi) with the intent of potentiating the genotoxic effects of DNA damaging agents such as chemo- and radiotherapy. However, preclinical studies rapidly uncovered tumor-specific cytotoxicity of PARPi in a subset of cancers carrying mutations in the *BReast CAncer 1* and *2* genes (*BRCA1/2*), which are defective in the homologous recombination (HR) DNA repair pathway, and several PARPi are now FDA-approved for single agent treatment in *BRCA*-mutated tumors. This phenomenon, termed synthetic lethality, has now been demonstrated in tumors harboring a number of repair gene mutations that produce a BRCA-like impairment of HR (also known as a ‘BRCAness’ phenotype). However, *BRCA* mutations or BRCAness is present in only a small subset of cancers, limiting PARPi therapeutic utility. Fortunately, it is now increasingly recognized that many small molecule agents, targeting a variety of molecular pathways, can induce therapeutic BRCAness as a downstream effect of activity. This review will discuss the potential for targeting a broad range of molecular pathways to therapeutically induce BRCAness and PARPi synthetic lethality.

## 1. Introduction

Poly(ADP-ribose) polymerase (PARP) proteins catalyze the transfer of an ADP (adenosine diphosphate)-ribose subunit of nicotinamide adenine dinucleotide (NAD^+^) onto a broad range of proteins, forming poly(ADP-ribose) (PAR) polymers. PARP1, which is responsible for the majority of cellular PARylation [1], is activated when its N-terminal DNA binding domain recognizes and binds to damaged DNA structures [2]. The recognition of single strand breaks (SSBs) by PARP1, and its subsequent auto-PARylation, mediates interaction with the molecular scaffold protein XRCC1 and the coordination of the transient assembly of multiprotein complexes that perform the post-recognition steps of SSB repair. Simultaneously, PARylation of histone proteins around the damage site remodels the surrounding chromatin to allow repair [3,4]. When the PARP1 protein level is reduced, either by CRISPR knockout [5] or small interfering RNA (siRNA) targeting [6], or catalytic activity is impaired, by expression of a catalytically inactive mutant [7] or targeting by small molecule inhibitors [8], a delayed repair of SSBs is observed leading to G2/M arrest, chromosomal instability, and cytotoxicity.

Observations that analogs of the PARP1 catalytic byproduct nicotinamide inhibit PAR synthesis in vitro [9] provided the template for the development of small molecule PARP inhibitors (PARPi). PARPi that have now received FDA approval include olaparib (AstraZeneca/Merck/KuDOS, Cambridge, UK; approved 2014), rucaparib (Clovis, Boulder, CO, USA; approved 2016), and niraparib (GlaxoSmithKline, Brentford, UK; approved 2017), while additional agents veliparib (AbbVie, North Chicago, IL, USA) and pamiparib (Beigene, Beijing, China) are currently being evaluated in phase III trials. In 2018, the second generation PARPi talazoparib (Pfizer, New York, NY, USA) was approved by the FDA, offering enhanced potency over its predecessors [10]. PARPi interact with the binding site of the PARP substrate NAD^+^, reducing PARylation activity and hence impairing single strand break repair (SSBR) capacity. It is now recognized that each of the clinically available PARPi possess differing abilities to ‘trap’ PARP at damage sites [11,12], with talazoparib possessing a trapping capacity 100-fold greater than the next most potent PARP trapper, niraparib.

PARPi selectivity for cancer cells harboring defects in DNA double strand break (DSB) repair was first described in concurrent publications by Farmer et al. [13] and Bryant et al. [14] in 2005, with both groups reporting exquisite PARPi sensitivity in *BRCA*-mutated tumors. BRCA1 and BRCA2 proteins play central roles in homologous recombination, the high-fidelity pathway responsible for the repair of DSBs during DNA replication [15]. Importantly, defects in HR stemming from *BRCA* mutations increase susceptibility to single-stranded base damage. Single-strand interruptions or adducts block replication fork progression, leading to extended fork stalling or collapse into a DSB–both of which require the proficiency of HR mechanisms for successful resolution [16]. Small molecule inhibitors of PARP1 are potent inducers of replication fork disruption, whether by the inactivation of the catalytic signaling that prevents SSBR, or by direct blocking of fork progression through the trapping of PARP into the DNA. Failure of *BRCA*-mutant cells to repair these PARPi-induced lesions is associated with an accumulation of cytotoxic DSB damage and the eventual activation of apoptotic mechanisms to limit the deleterious effects of error-prone repair [13,14].

The discovery that *BRCA* mutation and PARP inhibition, two independently non-lethal defects, combine to induce potent cell death forms the prototypical example of synthetic lethality [17]. This powerful concept allows cancer-specific defects to inform an effective therapeutic strategy with few off-target complications. However, *BRCA* mutations are relatively rare, associated with 5–10% of breast and ovarian cancers (and a lower percentage in other tumor types). As such, there is a growing push for the identification of tumors that exhibit ‘BRCAness’, mimicking the defective HR phenotype of *BRCA* mutation and potentially expanding the therapeutic utility of PARPi to a larger subset of tumors [18,19]. The BRCAness phenotype has been linked to mutations in other repair factors involved in DNA damage response (such as *ATM*, *ATR*, *RAD51*, or the Fanconi Anemia (FA) family), as well as altered gene expression by epigenetic silencing or cellular regulatory mechanisms [20]. To capitalize upon the potential of PARPi treatment in these settings, a number of methods have been explored to detect tumor BRCAness and predict PARPi sensitivity, including panel sequencing for DNA repair gene mutations, repair gene expression microarrays, testing for surrogate markers of HR deficiency such as loss of heterozygosity or sequence deletions associated with junctional microhomology, or functional tests of repair capacity such as RAD51 foci formation (recently reviewed in [21]). However, despite the translation of several of these methods into clinical trials, the predictive value of such tests in the context of PARPi therapy has not yet been conclusively established [22].

More recently, it has become recognized that BRCAness may be induced using therapeutic agents that modulate a variety of molecular pathways, potentially providing a novel method for inducing synthetic lethality in cancers that are otherwise HR-proficient. This review will focus on the therapeutic targeting of major molecular pathways that have been implicated in BRCAness, including epigenetic mechanisms, cell cycle checkpoints, and receptor kinase activity.

## 2. Modulation of Epigenetic Pathways

Epigenetics refers to heritable factors that influence cellular phenotypes other than DNA sequence, including DNA methylation, histone modification, and gene silencing by non-coding RNAs [23]. Inhibitors of the first two of these mechanisms have been linked to the induction of BRCAness (Figure 1).

### 2.1. Inhibition of DNA Methylation

DNA methyltransferases (DNMTs) are critical mediators of epigenetic gene regulation, responsible for genome-wide de novo and maintenance methylation. Aberrations in methylation have been widely implicated in cancer development, progression, and response to treatment [24,25], and consequently, DNMT inhibitors (DNMTi) including decitabine and 5-azacytidine have been developed and are now FDA-approved for the treatment of myelodysplastic syndrome [26,27]. These agents are cytosine analogs that become incorporated into replicating DNA, where they are targeted for methylation by DNMTs. Due to their altered structure, they cannot be released by DNMT by β-elimination, leading to the covalent entrapment of DNMT into the DNA [28,29].

There is evidence of a biological interplay between DNMT1, the enzyme responsible for maintenance methylation, and PARP1, that provides a rationale for combination DNMTi-PARPi therapy. DNMT1 and PARP1 are members of a multiprotein complex that localizes to sites of oxidative DNA damage [30], where the presence of PARylated PARP1 inhibits methylation activity by DNMT1 [31], possibly to maintain an open chromatin structure to permit repair. Our group has demonstrated that combining low-dose DNMTi treatment with the potent PARP-trapping PARPi talazoparib enhances PARP1-DNA binding, synergistically enhancing cytotoxicity across a number of *BRCA*-wildtype cancer types with minimal toxicity in in vivo models [32,33,34] or human subjects [35,36]. Similar synergism has also been observed when talazoparib is combined with the second-generation DNMTi guadecitabine [37].

In addition to a direct reduction in the free enzyme pool, DNMT entrapment also induces ubiquitin-E3 ligase-mediated proteasomal degradation of free DNMT1 [38,39]. Accordingly, low doses of DNMTi are sufficient to alter methylation patterns across the genome, leading to widespread alterations in multiple molecular pathways including the DNA damage response (DDR) and apoptosis [40]. Among the myriad of pathways contributing to the Hanahan and Weinberg ‘hallmarks of cancer’ [41] that are altered by DNMTi, our recent examination of the DNA repair reactome in non-small cell lung [33], breast, and ovarian cancers [34] demonstrated a significant reduction in DSB repair, particularly involving the FA pathway. Of note was the downregulation of FANCD2, which is mono-ubiquitinated by other FA pathway members in response to DNA damage, leading to colocalization with BRCA1 and BRCA2 during homologous recombination repair of DSBs, and resulting in it being ascribed a role as a BRCAness gene [42]. Furthermore, FANCD2 monoubiquitination is required for interactions with FANCD2/FANCI-associated nuclease 1 (FAN1), which mediates the canonical FA roles of interstrand crosslink repair and the resolution of stalled replication forks, potentially including those induced by trapped PARP1 and/or DNMT1 [43]. In keeping with the loss of these repair roles, DNMTi-induced FANCD2 downregulation was associated with a BRCAness phenotype, including increased replication fork stalling, DSB accumulation as measured by γH2AX foci accumulation, and a reduction in RAD51-mediated DSB repair capacity. Accordingly, in several human cancer cell lines and murine xenograft models, combining a low dose DNMTi with the PARPi talazoparib produces a significant and synergistic increase in tumor cell cytotoxicity [33,34]. These results have led to a dose-finding Phase 1 trial in untreated or relapsed/refractory AML using DNMTi decitabine and PARPi talazoparib [44], and a Phase 1 trial in BRCA-proficient breast cancer treated using oral decitabine and talazoparib (Table 1).

DNMTi-induced reversal of cancer-associated methylation abnormalities can reactivate abnormally methylated tumor suppressor gene promoters. One emerging example is Schlafen 11 (SLFN11), which irreversibly inhibits replication in cells undergoing replication stress such as DNA-damaging chemotherapy [45]. High levels of SLFN11 destabilizes the interaction between single-stranded DNA and replication protein A (RPA) at the sites of DNA damage, inhibiting downstream DSB repair and producing cell cycle checkpoint activation [46]. The suppression of SLFN11 expression is observed in ~50% of cancer cell lines and is correlated with a resistance to DNA-damaging agents including PARP inhibitors [47]. SLFN11 suppression appears to be primarily epigenetic in origin, linked to promoter methylation, histone deacetylation, and PRC-mediated histone methylation [45]. Decitabine can reverse *SLFN11* promoter methylation, leading to the re-expression and re-sensitization to DNA-damaging agents, and similar results have also been observed following EZH2 [48] or HDAC inhibitors [49] (see below), providing a further rationale for future studies combining epigenetic agents with PARP inhibitors.

### 2.2. Maintenance of Chromatin Repressive States

Histone deacetylases (HDACs) remove acetyl groups from ϵ-N-acetyl lysine residues on histones, leading to chromatin condensation and transcriptional repression. Abnormal acetylation resulting from HDAC overexpression can downregulate the expression of various tumor suppressive mechanisms, including cyclin-dependent kinases, differentiation factors, and proapoptotic signals, leading to the uncontrolled proliferation, de-differentiation, and survival that is characteristic of oncogenesis and metastasis [50]. The eighteen identified members of the HDAC family have been classified into four groups (class I, IIa/b, V, and III/sirtuins) based on homology to yeast HDACs. Compounds with anti-HDAC activity are numerous, and can be divided into pan-HDAC inhibitors (HDACi), which exhibit activity against all non-sirtuin HDACs, or selective HDACi, which target specific HDACs [51]. FDA approval has been granted for the treatment of various hematological malignancies for three pan-HDACi (vorinostat, belinostat, and panobinostat) and one HDAC1/2-selective HDACi (romidepsin) [52].

Acetylation exerts effects over the chromatin structure that impacts the recognition and repair of DNA damage [53], and accordingly, HDACi have been reported to alter DSB repair capacity [54]. While deacetylation activity of HDAC1/2 has been shown to both directly and indirectly decrease c-NHEJ activity [55,56], the role of HDACs in HR, and hence the therapeutic potential of HDACi for the induction of BRCAness, is less clearly defined. Of note, HR proteins including BRCA1, BRCA2, and RAD51 have been reported to be suppressed by HDACi in a variety of cancers [57,58], sensitizing to PARPi [59,60,61,62,63,64,65]. Based on these results, phase I trials combining olaparib with vorinostat are underway in advanced lymphoma and breast cancer (Table 1).

Notably, the inhibition of the deacetylation activity following HDACi exposure leads to PARP1 hyperacetylation and enrichment in chromatin that resembles PARPi-induced PARP trapping. When combined with PARPi, HDACi treatment further increases PARP trapping, synergistically sensitizing to the PARP-trapping PARPi talazoparib [66]. Synergism has also been observed when HDACi are combined with DNMTi, specifically by enhancing the re-expression of genes silenced by abnormal promoter methylation [35,67]. Valdez et al. have reported synergistic inhibition of AML and lymphoma cell proliferation by the triple combination of PARPi niraparib, DNMTi decitabine, and HDACi romidepsin or pabinostat, associated with the activation of ATM-mediated DDR, increased ROS production, and the induction of apoptosis [68]. These effects were hypothesized to be the sequelae of DSB accumulation induced by triple combination through three mechanisms: significantly enhanced PARP trapping; acetylation and inhibition of DNA repair proteins including Ku70/80 and PARP1; and the downregulation of the nucleosome-remodeling deacetylase complex, a transcriptional repressor with chromatin remodeling activity that is functionally linked to efficient DNA repair [69]. While further preclinical study is required, these results provide a rationale for the future development of combination therapy using PARPi, HDACi, and DNMTi.

### 2.3. Polycomb Repressive Complex 2

An enhancer of the zeste homolog 2 (EZH2) is the histone methyltransferase subunit of polycomb repressive complex 2 (PRC2), which methylates histone H3 on lysine 27 (H3K27me3) to mark chromatin as transcriptionally silent. PRC2 plays an important oncogenic role through the modulation of the DDR [70]. EZH2 overexpression, which is common in many cancers [71], induces the downregulation of RAD51 homolog expression [72], cytoplasmic BRCA1 retention [73], and impaired HR that is associated with increased genomic instability. PRC2 appears to play a role in the DSB repair pathway choice, being recruited to DSBs in a Ku-dependent mechanism to promote efficient NHEJ [74], and accordingly, EZH2 depletion favors HR, impairs NHEJ, and sensitizes to irradiation damage [75]. Recent evidence indicates that this DSB repair pathway switch can be therapeutically targeted by PARPi in a subset of HR-proficient tumors overexpressing the oncogene coactivator-associated arginine methyltransferase 1 (CARM1). The overexpression of CARM1 promotes the EZH2 silencing of *MAD2L2*, a member of the shieldin complex that limits DNA end resection to favor NHEJ. Accordingly, in CARM1-high cells, EZH2 inhibition upregulates *MAD2L2*, increasing error-prone NHEJ activity and associated chromosomal abnormalities, and producing mitotic catastrophe in combination with PARPi treatment [76]. An ongoing phase II clinical trial, evaluating multiple targeted therapies in a biomarker-guided precision therapy approach, includes the novel agents SHR2554 (EZH2 inhibitor) and SHR3162 (PARPi) (Table 1).

### 2.4. BET Proteins

The conserved bromodomain and extraterminal (BET) family of proteins are characterized by two tandem bromodomains that bind to activated lysine residues on target proteins [77,78]. BET members preferentially interact with hyperacetylated histones, leading to an accumulation at the transcriptionally active regulatory elements [79]. BET family member BRD4 acts as a transcriptional cofactor, influencing the expression of a wide range of genes involved in cell fate determination. In cancer, BRD4 has been implicated in the activation of a multitude of oncogenes, co-occupying a set of promoter super-enhancers associated with prominent oncogenic drivers such as c-MYC [80,81]. High affinity small molecules targeting the BET bromodomains demonstrate preclinical efficacy in a wide range of cancers associated with transcriptional suppression of key proto-oncogenes including *c-MYC*, *N-MYC*, *FOSL1*, and *BLC2* (reviewed in [79].

The first report of potential synergism between BET inhibition and PARPi was based on a drug combination screen testing PARPi olaparib in *BRCA*-wildtype triple negative breast (TNBC), ovarian, and prostate cancer in combination with 20 well-characterized epigenetic modulators across seven classes, demonstrating synergism for all tested BET inhibitors (BETi) [82]. BETi treatment significantly enhanced PARPi-induced DSB accumulation independent of PARP-trapping, associated with the repression of *BRCA1* and *RAD51* transcription, which is suggestive of induced BRCAness. Notably, BETi treatment could disrupt the enrichment of BRD2/3/4 at the *BRCA1* and *RAD51* promoter regions, in addition to the putative super-enhancer region downstream of the *BRCA1* promoter that exerted a stronger transcriptional enhancing activity than the promoter region alone [82]. Validation of these results, with similar BETi-induced repression of *BRCA1* and *RAD51*, induction of an HR defect, and sensitivity to PARPi, has since been reported in TNBC [83].

A subsequent study used publicly available transcriptional profiling data to demonstrate that BRD4 inhibition modulates a previously validated HR defect gene signature [84], though finding minimal impact on *BRCA1* or *RAD51* expression in cell lines of multiple cancer types. Instead, a consistent and marked downregulation of CtIP was observed, in keeping with ChIP-seq analysis that indicates that both the *CtIP* promoter and an associated enhancer region are directly targeted by BRD4. BETi induced PARPi sensitivity in 40 of 55 cancer cell lines and five in vivo models spanning breast, ovarian, and pancreatic cancer, as well as resensitizing PARPi-resistant cells [85]. Despite the mechanistic discrepancies between the studies, these results indicate the therapeutic potential of the PARPi-BETi combination that warrants further investigation.

## 3. Cell Cycle Checkpoints and the DNA Damage Response

Cell cycle progression is directed by the activity of cyclin-dependent kinases (CDKs), which phosphorylate targets that include transcription factors and regulatory elements such as retinoblastoma (Rb), promoting checkpoint transit. Cell cycle arrest occurs in response to DNA damage, initiated by the activation of ataxia-telangiectasia mutated (ATM; by DSBs) and ataxia-telangiectasia and Rad3-related (ATR; by single strand breaks and stalled replication forks), leading to the phosphorylation of checkpoint kinases (CHK) 2 and 1. Activated CHK1 and CHK2 antagonize the function of the Cdc25 phosphatase family, allowing the accumulation of inhibitory phosphorylation on CDKs, thus delaying cell cycle progression so that DNA repair can occur [87] (Figure 2).

### 3.1. Cyclin-Dependent Kinases

CDKs are a family of proline-directed Ser/Thr kinases, first characterized for their highly evolutionarily conserved function in the cell cycle, although now recognized to also have an important role in the modulation of transcription [88]. CDK activity is promoted by conformational changes induced by binding to cyclin subunits, while Thr14/Tyr15 phosphorylation by regulatory proteins such as WEE1 inhibit CDK kinase function until removed by Cdc25 family phosphatases [88]. CDK1, which has roles in S and G2 phase transit, is required for the efficient recruitment of BRCA1 to DNA damage sites, with depletion or small molecule inhibition abrogating S phase checkpoint arrest [89], reducing BRCA1 recruitment [90], and impairing RPA and RAD51 loading [91]. Accordingly, CDK1 inhibition sensitizes to PARPi, both in vitro [92] and in *BRCA*-wildtype xenograft models [90].

Disruption of transcription-related CDK activity may also impact HR activity. CDK12, in complex with cyclin K, phosphorylates Ser2 at the C-terminal domain of RNA PolII, stimulating productive transcriptional elongation. Additionally, CDK12 phosphorylates and regulates pre-mRNA processing factors such as the U1 snRNP complex, which recognizes and inhibits intronic polyadenylation sites. The mutation of CDK12 or inhibition using the CDK12/13 inhibitor THZ531 produces premature cleavage and polyadenylation (PCPA), leading to truncated gene products, particularly in long (>45 kb) genes [93,94]. Notably, several DNA damage response genes are susceptible to PCPA due to their relatively long length and high number of cryptic polyadenylation sites, including *BRCA1*, *ATR*, and the FA genes *FANCI* and *FANCD2*, and hence CDK12 has been ascribed a role in the promotion of HR activity and genomic stability [95]. SR-4835, a novel selective CDK12/13 inhibitor, triggers PCPA at polyadenylation sites, impairing HR gene expression and efficiency, and sensitizing to PARPi [96]. Similarly, the multi-CDK targeting inhibitor dinaciclib reduces HR gene expression in TNBC specifically through the inhibition of CDK12, sensitizing the *BRCA*-wildtype in vitro and in vivo models to PARPi, and reversing de novo and acquired PARPi resistance in *BRCA*-mutated cells [97]. However, this preclinical promise has yet been translated into clinical trials.

A novel mechanism by which CDKs with transcriptional regulatory activity may impact gene expression is through interplay with epigenetic mechanisms. In euchromatin, CDK9 promotes transcriptional elongation by promoting RNA PolII promoter-proximal pause release [98]. Recent evidence suggests that it also plays a contrasting role in gene silencing, mediated through the regulatory phosphorylation of the SWI/SNF member BRG1, an ATP-dependent helicase that maintains heterochromatic DNA in a condensed conformation. Inhibition of CDK9 by the novel targeted agent MC180295 promotes dephosphorylation of BRG1 leading to chromatin opening and gene reactivation [99]. Amongst the epigenetically silenced genes reactivated by CDK9 inhibition are endogenous retrovirus (ERV) elements, which form cytoplasmic double-stranded RNA aggregations that trigger cellular immune signaling—the basis for the ‘pathogen mimicry’ immune response that has also been described following DNMTi treatment [35,36]. Our work in breast and ovarian cancer has recently linked ERV re-expression and immune signaling directly to the induction of HR defects, providing a mechanistic basis for the observed PARPi sensitivity both in DNMTi treatment [34] and following CDK9 inhibition [100].

### 3.2. ATR Inhibitors

ATR plays an important role in DNA damage sensing and the cellular response, particularly during S phase. As a member of the phosphatidylinositol-3-kinase-like kinase (PIKK) family, ATR directly phosphorylates Ser/Thr-Glu motifs on hundreds of target proteins across multiple processes involved in the DDR, in addition to activating CHK1-mediated kinase activity and cell cycle arrest [87]. ATR interacts with replication protein A (RPA), which coats single-stranded DNA during replication, allowing it to sense stalled replication forks [101], and it also interacts with ATM on resected ends of DSBs to promote repair [102,103]. Accordingly, ATR inhibition attenuates G2/M arrest following DNA damage, limits RAD51 focus formation, promotes early mitotic entry, and sensitizes to both DNA-damaging cytotoxic agents and PARPi [104].

ATR has been ascribed a role in the development of PARPi resistance in *BRCA1*-mutated cells, which can become ‘rewired’ to reverse the BRCAness phenotype. In *BRCA1*-mutant cells, the absence of BRCA1 recruitment to DSB ends is associated with the suppression of end resection, and the impaired recruitment of PALB2-BRCA2, which is required to load RAD51 onto resected single-stranded DNA ends to enact repair. Phosphorylation of RPA by ATR can bypass this block to HR by promoting PALB2-BRCA2 recruitment and subsequent RAD51 loading, reversing *BRCA1* mutation-associated HR deficiency [105]. Furthermore, ATR-mediated RAD51 recruitment plays an important role in the stabilization of stalled replication forks in the absence of BRCA1 [105,106], which ordinarily functions to prevent extensive resection by MRE11 [107]. These ATR-dependent pathways of reactivated HR and fork protection produce PARPi resistance that can be abrogated by ATR knockdown or small molecule inhibitors (ATRi) [108]. Several clinical trials combining the ATRi AZD6738 (ceralasertib, AstraZeneca, Cambridge, UK) and olaparib are underway in PARPi-naïve and PARPi-treated cancers (Table 2).

### 3.3. CHK1 Inhibitors

Activated by direct interaction with ATR, CHK1 signaling plays an important role in the cell cycle signaling pathway in response to replication stress. Notably, potent CHK1 activation is observed in *BRCA*-mutant cells treated with PARPi, reflecting an increased dependence upon ATR-mediated fork protection [108]. Akin to ATRi treatment, CHK1 inhibitors (CHK1i) induce unrepaired DSB DNA damage while releasing cells from cell cycle arrest into early mitosis [109]. Furthermore, CHK1 directly phosphorylates RAD51 at Thr309, stimulating recruitment after DNA damage [110] and suppressing proteasomal degradation [111,112]. However, despite inducing early mitotic entry, CHK1 inhibition does not produce the marked accumulation of DNA damage and robust PARPi sensitization observed with ATRi [108,113,114]. Accordingly, clinical trials of a CHK1i and PARPi combination have not been initiated at the same rate, with a single phase I of the CHK1i prexasertib in combination with olaparib being recorded on ClinicalTrials.gov (Table 2).

### 3.4. WEE1 Inhibitors

WEE1 is a member of the serine-threonine (Ser/Thr) kinase family that limits G2 progression by inactivating the phosphorylation of CDK1, thus cooperating with the inhibitory phosphorylation of the Cdc25 phosphatase by ATR/CHK1 [115]. Furthermore, activated WEE1 sustains ATR and CHK1 phosphorylation during the DDR to delay mitotic entry [116]. The inhibition of WEE1 results in the forced activation of CDK1, leading to the phosphorylation of BRCA2 that limits HR [117,118]. Similar to ATR, WEE1 has also been implicated in replication fork protection through direct interaction and negative regulation of DNA cleavage by the endonuclease MUS81 [119], which has structure specific activity against Holliday junctions formed during HR [120]. In keeping with these functions, the inhibition of WEE1 slows replication fork progression, limits HR, activates DDR, and forces mitotic entry, particularly in response to genotoxic agents [121]. Accordingly, the WEE1 inhibitor AZD1775 sensitizes gastric [122], non-small cell lung cancer (NSCLC) [123], and pancreatic cancer [124] to olaparib, particularly when combined with genotoxic stress via ionizing radiation, and a small number of clinical trials are currently underway (Table 2).

## 4. Tyrosine Kinase Signaling Pathways

### 4.1. FMS-Like Tyrosine Kinase 3 (FLT3)

FLT3 is a transmembrane receptor that plays a role in hematopoietic cell differentiation, proliferation, and survival via the phosphorylation of downstream targets including phosphoinositide 3-kinase (PI3K), protein kinase B (also known as AKT), Ras, mitogen-activated protein kinase (MAPK), and signal transducer and activator of transcription 5 (STAT5) [125] (Figure 3). FLT3 is predominantly expressed on CD34-positive hematopoietic stem or progenitor cells, where it drives early myeloid and lymphoid lineage development. Approximately 30% of acute myeloid leukemias (AML) carry poor prognosis FLT3 mutations, most commonly internal tandem duplications (ITD) that constitutively activate kinase activity and aberrant signaling via STAT5 to promote proliferation and survival [125,126]. Multitargeted tyrosine kinase inhibitors (TKIs) such as sunitinib and sorafenib possess activity against FLT3, but have demonstrated limited antileukemic activity in clinical trials, prompting the development of inhibitors with greater FLT3 specificity and potency such as gilteritinib and quizartinib [125]. Remissions associated with FLT3 inhibition are usually short-lived, commonly due to the persistence of therapy-refractory leukemia stem cells (LSCs), despite clearance of the bulk of leukemia progenitor cells (LPCs) [127]. Therefore, new approaches are required to enhance therapeutic targeting of LSCs and prolong the efficacy of these agents.

AML cells generate an elevated level of reactive oxygen species (ROS) that has been linked to STAT5 activation of RAC1, a critical member of the NADPH oxidase (NOX) pathway [128,129]. This ROS generation is associated with an increased DNA damage burden, including DSBs [129], that is frequently accompanied by the functional dysregulation of DSB repair [130]. In certain subsets of AML, such as those driven by the fusion oncoproteins AML1-ETO or PML-RARα, the expression of key HR proteins (including RAD51, ATM, BRCA1, and BRCA2) is suppressed, producing sensitivity to PARP inhibition [131]. In contrast, DSB repair is proficient in FLT3-ITD^high^ leukemic cells [132,133], rendering resistance to PARPi [131]. Treating FLT3-ITD^high^ cells with quizartinib suppresses downstream signaling by Janus kinase 2 (JAK2) and PI3K to produce a rapid downregulation of DSB repair proteins (including BRCA1, BRCA2, PALB2, and RAD51) that sensitizes to PARP inhibition in preclinical and in vivo models [134]. Importantly, this effect is observed in both LPCs and in proliferating and quiescent LSCs, including those cultured under conditions mimicking the bone marrow microenvironment, an important reservoir of treatment-resistant cells [134]. Of note, these effects may be mediated in part by the effects of PARP1 inhibition outside of DNA repair, such as STAT5 protein stabilization through PARylation [135]. This approach may therefore offer a potential to improve long-term outcomes by targeting the disease-initiating stem cell population without losing sensitivity in the presence of resistance mutations.

### 4.2. c-MET

The c-MET transmembrane receptor tyrosine kinase is expressed on stem and progenitor cells, where activation by its ligand hepatocyte growth factor (HGF) activates an invasive growth program during embryogenesis and organ regeneration [136]. C-MET activation is a common event in numerous cancers, driving a number of proliferation, differentiation, and survival signaling pathways, including sustained Ras/MAPK activity, as well as PI3K, STAT, β-catenin, NFκB, and Notch activation [137] (Figure 3). Several small molecules and antibodies targeting the HGF-MET pathway have been developed, categorized as ligand inhibitors (blocking pro-HGF cleavage to the active form or preventing ligand-receptor binding) or MET receptor inhibitors (competitively antagonizing receptor binding or inhibiting MET tyrosine kinase activity). Of these, TKIs with a specific activity against MET kinase (capmatinib) or non-specific global kinase inhibition (crizotinib, cabozantinib) have progressed farthest in clinical development, with FDA approval for indications such as NSCLC, renal cell, hepatocellular, and medullary thyroid cancers [137].

In addition to the activation of signaling cascades, c-MET interacts and phosphorylates a wide range of target proteins. Recent evidence indicates that c-MET becomes activated following oxidative stress, inducing an antiapoptotic cytoprotective response that includes the phosphorylation of PARP1 at Tyr907 located within the catalytic domain [138]. Phospho-Tyr907 not only enhances PARylation activity, but also reduces PARPi binding, and may be a predictive marker of PARPi resistance [139]. The inhibition of c-MET using the non-specific pan-kinase inhibitor crizotinib [139] or the MET-specific HS-10,241 [140] abolishes Tyr907 phosphorylation, sensitizing to PARPi in in vitro and xenograft models of TNBC, NSCLC, and high-grade serous ovarian cancer (HGSOC), independent of BRCA status. Notably, elevated c-MET expression in *BRCA*-mutant TNBC cell lines correlates to PARPi resistance that can be reversed by c-MET inhibition. These results highlight the potential of a therapeutic strategy to combine PARPi with c-MET inhibitors in PARPi-resistant cancers.

### 4.3. EGFR

The epidermal growth factor receptor (EGFR) is a transmembrane receptor tyrosine kinase that binds to a variety of ligands including epidermal growth factor (EGF) and transforming growth factor α (TGFα) [141]. Upon ligand binding, inactive EGFR monomers dimerize to an active form, either as homodimers or as heterodimers with other members of the ErbB receptor family such as HER2. Autophosphorylation induces the activation of target proteins via interaction at phosphotyrosine SH2 domains, resulting in the stimulation of signal transduction cascades including MAPK (see below), AKT, and JNK (Figure 3). Ultimately, EGFR activation stimulates a phenotype that promotes proliferation, cell adhesion, and migration; hence, constitutive activation is an oncogenic driver in multiple human cancers. TKIs with specific activity against EGFR (erlotinib, gefitinib, and the EGFR/HER2-targeting lapatinib) and monoclonal antibodies that prevent EGFR-ligand binding (cetuximab and panitumumab) have been approved for use in a variety of EGFR-expressing malignancies, including NSCLC, head and neck squamous cell carcinoma (HNSCC), and colorectal cancer.

In response to EGFR inhibition, physical interaction between EGFR and components of the classical NHEJ pathway of DSB repair (particularly DNA-PK_CS_) have been reported [142], leading to globally reduced levels of DNA-PK [143,144] and/or subcellular relocalization away from the nucleus [145] that reduces NHEJ repair capacity and sensitizes to radiation [146]. EGFR inhibition has also been linked to the transient downregulation of mismatch repair (*MLH1*, *MSH2*, and *MSH6*) and HR (*BRCA2*, *RAD51*) genes in cetuximab-sensitive colorectal cancer cell lines, increasing the mutability that leads to the development of permanent resistance with prolonged exposure [147]. Although a potential strategy to target this repair downregulation has not yet been extensively explored, the Hung lab have built upon their work combining MET inhibitors with PARPi (see above) to demonstrate that EGFR cooperates with MET in subsets of hepatocellular cancers [148] and TNBCs [149] to phosphorylate PARP1 Tyr907 in response to DNA damage, demonstrating that dual EGFR/MET inhibition is required in this group to block phosphorylation and sensitize resistant cells to PARPi. This may further broaden the therapeutic potential of MET inhibition to overcome PARPi resistance in certain cancers.

### 4.4. VEGFR

Vascular endothelial growth factor receptors (VEGFRs) are tyrosine kinase receptors that play critical roles in signal transduction during vasculogenesis and angiogenesis [150]. The abnormal expression of VEGFR ligands (VEGFs) by tumor-associated macrophages contributes to tumor neoangiogenesis, an observation that led to the development of targeted anti-angiogenic therapies. An important consequence of VEGFR inhibition is tumor hypoxia [151], a state that has been linked to HR defects via the downregulation of BRCA1, BRCA2, and RAD51 [152,153]. Accordingly, in preclinical models, VEGFR inhibition is reported to sensitize to PARPi, and a phase 2 clinical trial in platinum-sensitive HGSOC indicates that the combination prolongs progression-free survival over single agent treatment [154]. Several early-stage clinical trials combining the VEGFR inhibitor cediranib with olaparib are currently underway (Table 3).

### 4.5. MAPK Pathway

The mitogen-activated protein kinase (MAPK) signaling pathway regulates a diverse range of cellular processes, including proliferation, differentiation, and survival [155]. In response to extracellular stimuli including growth factors or cytokines, a transmembrane receptor-linked tyrosine kinase (such as EGFR, fibroblast growth factor receptor [FGFR], or FLT3) activates a member of the Ras subfamily. This triggers a kinase signaling cascade through RAF, MEK1/2, and ERK (also known as MAPK) (Figure 3), leading to the phosphorylation of a range of target proteins including transcription factors (such as c-MYC, c-Jun, and c-Fos), cell cycle proteins (such as CDK4/6 for S-phase entry), apoptotic factors (inactivating pro-apoptotic proteins such as Bad, Bim, and caspase 9), and regulators of the translational machinery such as the 90 kDa ribosomal S6 kinase (p90^Rsk^) [155]. Activated Ras can also interact with other signaling pathways, such as the Pi3K-AKT-mTOR (mammalian target of rapamycin) pathway (see below) and RAS-like (RAL) GTPases. Unsurprisingly, members of the MAPK pathway are proto-oncogenes, with constitutive activation of Ras observed in ~30% of cancers, and BRAF in ~7% [156]. While Ras has not proven to be an effectively druggable target, inhibitors of downstream MAPK members such as MEK are under development.

Recent evidence indicates that constitutive Ras activation produces a PARPi-resistant phenotype. Using a reverse phase protein assay to determine changes in signaling pathway expression, Sun et al. demonstrated that transient PARPi treatment induces Ras/MAPK activation, producing a downregulation of the pro-apoptotic targets that induced PARPi resistance, and furthermore, recapitulated the PARPi resistance observed in *Ras*-mutant cell lines. MEK (or ERK) inhibition in *KRAS*-mutant or *KRAS-*induced cell lines enforced the expression of the Ras-regulated FOXO3a transcription factor, leading to increased expression of pro-apoptotic factors such as Bim. Phosphorylation patterns of multiple DNA repair proteins were also found to be altered by MEKi, associated with altered expression levels of DSB repair proteins that reversed an enhanced level of DSBR observed in *KRAS*-mutant cells. This reduced MEKi-induced reduction in DSBR sensitized *KRAS-*mutant cells to talazoparib, compounded by increased PARP1 expression that enhanced the accumulation of cytotoxic PARP-trapping lesions [157]. These results were subsequently confirmed by a second group [158], suggesting a combinatorial role for PARPi and MEKi in the treatment of PARPi-resistant and/or *KRAS-*mutant tumors that is now being tested in a phase I/II trial (Table 3). Interestingly, PARPi synergism is not recapitulated by BRAF inhibition, likely because other RAF homologs bypass the effects of therapeutic inhibition [157].

### 4.6. PI3K Pathway

The phosphoinositide 3-kinase pathway is a major effector of receptor tyrosine kinase activation, transducing signals via phospholipid generation to protein kinase B (also known as AKT), mammalian target of rapamycin (mTOR), and other downstream targets [159]. Loss of function mutations in the negative regulator phosphatase and tensin homolog (*PTEN*) are a common occurrence in human cancers, as are activating mutations of other components of the PI3K pathway, producing an accelerated growth and proliferation phenotype. PI3K has long been ascribed a role in promoting the DDR [160], including regulating the binding of the NBS1 damage sensor to DNA [161], and the control of RAD51 recruitment to DSBs [162]. The downstream effector mTOR also modulates the DDR, maintaining HR and NHEJ [163,164], at least in part by the stimulation of FANCD2 expression [165,166]. The PI3K pathway also exerts transcriptional control over repair gene expression, including *BRCA1/2*, *RAD51* [167,168], *PRKDC* (DNA-PK_CS)_, and *ATM* [169]. Several studies have therefore considered the potential for PI3K pathway inhibitors to induce DSB repair defects. In both *in vitro* and *in vivo* models of BRCA-proficient TNBC, BRCA1/2 downregulation induced by the PI3K inhibitor (PI3Ki) BKM120 [167], mTOR inhibitors everolimus or KU0063794 [170], or the dual PI3K/mTOR inhibitor GDC-0980 [171] impairs HR and sensitizes to PARPi. Similar results have been observed in *PTEN*-mutant, PI3K-activated endometrial cancer [172], and in *PI3K*-wildtype [173,174] or mutant [175] ovarian cancer. Phase I/II clinical trials examining PARPi in combination with inhibitors of PI3K, AKT, or mTOR are underway (Table 3).

## 5. Other Targets

### 5.1. BCR-ABL

c-ABL tyrosine kinase is constitutively activated in most chronic myeloid leukemias (CML) following translocation adjacent to the BCR gene, forming the BCR-ABL ‘Philadelphia chromosome’ [176]. Activated c-ABL interacts with multiple proliferative and survival pathways, including MAPK, PI3K, and JAK/STAT. Following activation induced by IR, c-ABL phosphorylates RAD51 at Tyr315, enhancing complex formation with RAD52 [177,178] and stabilizing chromatin binding [179]. In the presence of BCR-ABL, RAD51 and RAD51 paralog expression is significantly enhanced, mediated via JAK/STAT signaling [180]. Imatinib is a multi-kinase inhibitor that possesses selectivity for BCR-ABL, along with c-kit and PDGFR, and is FDA-approved in hematological malignancies and gastrointestinal stromal tumors. Imatinib reduces RAD51 nuclear expression and chromatin binding, and inhibits HR-mediated repair [181], thus sensitizing to PARPi in ovarian cancer [174]. Further study, particularly in the BCR-ABL fusion setting, is required to evaluate the clinical potential of this combination.

### 5.2. NAMPT Inhibition

Nicotinamide phosphoribosyl transferase (NAMPT) is a rate-limiting enzyme required for the generation of the PARP substrate β-NAD^+^. Small molecule inhibition of NAMPT suppresses β-NAD^+^ synthesis, preventing PARP1 PARylation activity. Synthetic lethality between an experimental NAMPT inhibitor and olaparib has been observed in different tumor models independent of *BRCA* status, producing a synergistic and non-redundant NAD^+^ depletion, a reduction in PARylation, an increase in DNA damage, and an induction of apoptosis [182,183]. While this combination does not induce a BRCAness phenotype, it may offer an opportunity to further optimize therapeutic strategies by maximizing PARP inactivation.

### 5.3. Pharmacological Ascorbate

High doses of vitamin C (ascorbate) have been evaluated as an anticancer therapy in a range of malignancies. Cytotoxicity is mediated in part through DNA damage accumulation resulting from the generation of hydrogen peroxide, which activates PARP1 and subsequently depletes the PARP1 substrate nicotinamide adenine dinucleotide (NAD+) leading to ATP depletion and cell death [184]. Although PARPi treatment prevents NAD+/ATP depletion, cell death still ensues secondary to DSB accumulation linked to the ascorbate-induced downregulation of BRCA1, BRCA2, and RAD51 [185].

Additionally, low doses of vitamin C, particularly in the context of vitamin C deficiency, may synergistically enhance the effects of DNMTi in hematological malignancy [186]. Vitamin C acts as a cofactor for ten-eleven translocation (TET) enzymes that convert 5-methylcytosine (5mc) to 5-hydroxymethylcytosine (5 hmc), thus producing active demethylation in concert with the passive demethylation induced by DNMT inhibition. Notably, these demethylation effects are enriched at the long terminal repeat (LTR) regions of ERVs, leading to ERV re-expression and pathogen mimicry immune responses [187] that induce HR defects as previously described in Section 3.1. An early clinical trial has demonstrated an enhanced 5 hmc/5 mc ratio following oral vitamin C supplementation compared to placebos in patients with myeloid cancers treated with DNMTi [186], although the impact upon HR capacity and PARPi efficacy has not yet been considered.

## 6. Conclusions

PARP inhibitor sensitivity in *BRCA*-mutated breast and ovarian cancers is the prototypical example of synthetic lethality but represents only a small number of total cancer diagnoses. To expand PARPi utility into a wider setting, research has primarily focused on the identification of other genetic and epigenetic determinants of BRCAness. With the advent of increasing numbers and a widening scope of targeted therapies, it has become apparent that BRCAness—and hence PARPi sensitivity—may also be pharmaceutically induced. With the exploration of the additional roles of PARP in the regulation of gene expression and protein translation, this may increase the targets for the induction of BRCAness. This review has summarized the current status of therapeutic BRCAness induction, focusing on epigenetic agents, drugs targeting cell cycle checkpoints and the DNA damage response, and tyrosine kinase inhibitors. The translation of promising preclinical results to clinical trials and beyond is critical to maximizing PARPi therapeutic scope and optimizing treatment outcomes.

## Figures and Tables

**Figure 1 cancers-14-02640-f001:**
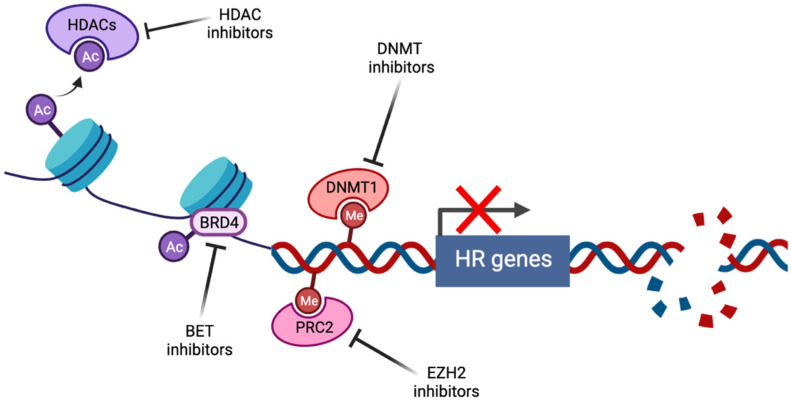
Induction of BRCAness by pharmacological targeting of epigenetic pathways (created with Biorender.com, accessed 15 May 2022). Histone deacetylase inhibitors (HDACi) prevent the histone deacetylation activity of HDACs, thus maintaining chromatin in a condensed state associated with transcriptional repression. Bromodomain and extraterminal (BET) family members such as BRD4 act as transcriptional cofactors at acetylated gene promoters, including the homologous recombination (HR) genes *RAD51* and *BRCA1*, whose expression is suppressed by BET inhibition. DNA methyltransferase (DNMT) and enhancer of zeste homolog 2 (EZH2) inhibitors alter genome-wide methylation patterns and have been linked to altered DNA double strand break repair gene expression. In each case, repression of HR gene expression and activity has been described, contributing to induction of the BRCAness phenotype.

**Figure 2 cancers-14-02640-f002:**
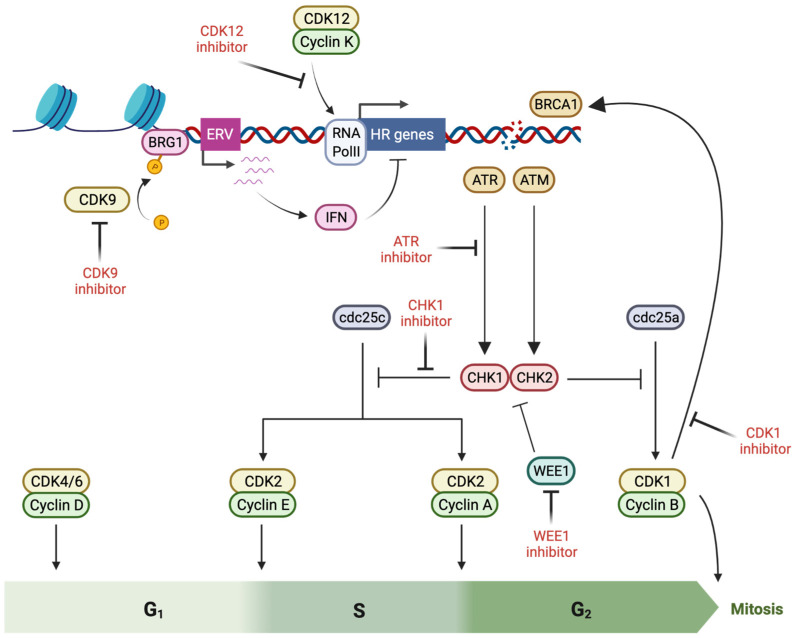
Induction of BRCAness by pharmacological targeting of cell cycle checkpoint proteins (created with Biorender.com, accessed 15 May 2022). Cell cycle arrest is initiated as a component of the response to DNA damage, initiated by activation of ATM (ataxia-telangiectasia mutated) and ATR (ataxia-telangiectasia and Rad3-related), which lead to CHK (checkpoint kinase) 2 and −1 phosphorylation and Cdc25 antagonism, producing inhibitory phosphorylation of cyclin-dependent kinases (CDK) that prevents cell cycle progression. Multiple factors in this cell cycle checkpoint response directly interact with double strand break repair proteins, promoting repair activity–and thus offering a potential for BRCAness induction via inhibitory molecules.

**Figure 3 cancers-14-02640-f003:**
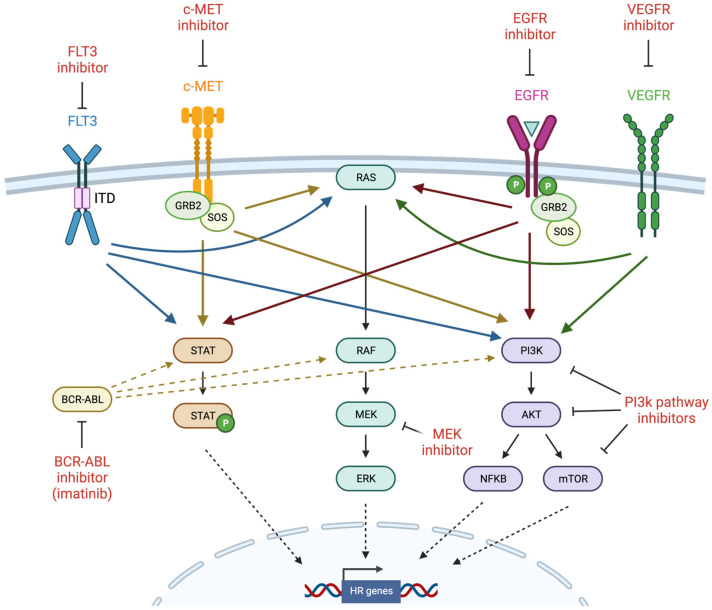
Induction of BRCAness by tyrosine kinase inhibitors (created with Biorender.com, accessed 15 May 2022). Receptor tyrosine kinases (RTKs) such as FLT3, c-MET, EGFR, and VEGFR are frequently mutated and constitutively activated in cancer, and thus are important targets for therapeutic inhibition. RTKs activate several major intracellular signaling pathways such as JAK/STAT, MAPK, and PI3K, which have also been the target of inhibitor development. Homologous recombination genes have been identified among the transcriptional targets of these signaling pathways, leading to investigation of TKIs as potential inducers of BRCAness. GRB2 = growth factor receptor bound protein 2; SOS = son of sevenless.

**Table 1 cancers-14-02640-t001:** Clinical trials evaluating epigenetic therapy in combination with PARPi.

ClinicalTrials.gov Identifier [86] (accessed 20 May 2022)	Phase	Epigenetic Drug	PARPi	Other Drugs	Cancer	Status
*DNMT inhibitor*						
NCT02878785	I/II	Decitabine	Talazoparib		Untreated or R/R ^1^ acute myeloid leukemia (AML)	Active, not recruiting
*HDAC inhibitor*						
NCT03259503	I	Vorinostat	Olaparib	Gemcitabine, busulphan, melphalan	R/R lymphoma undergoing stem cell transplant	Recruiting
NCT03742245	I	Vorinostat	Olaparib		R/R or metastatic breast	Recruiting
*EZH2 inhibitor*						
NCT04355858	II	SHR2554	SHR3162		Luminal advanced breast	Recruiting

^1^ R/R = relapsed/refractory.

**Table 2 cancers-14-02640-t002:** Clinical trials evaluating cell cycle inhibitors in combination with PARPi.

ClinicalTrials.gov Identifier [86] (accessed 20 May 2022)	Phase	Status	Cell Cycle Inhibitor	PARPi	Other Drugs	Cancer
*ATR* inhibitor						
NCT02264678	I/Ib	Recruiting	AZD6738	Olaparib	Carboplatin, durvalumab	Advanced solid tumors
NCT02576444	II	Active, not recruiting	AZD6738	Olaparib		Advanced solid tumors with *ATM*, *CHK2*, *MRN* mutation
NCT02723864	I	Recruiting	VX-970	Veliparib	Cisplatin	Advanced refractory solid tumors
NCT02937818	II	Active, not recruiting	AZD6738, AZD1775	Olaparib	Carboplatin	Platinum-refractory small cell lung cancer (SCLC)
NCT03182634	II	Recruiting	AZD6738	Olaparib		TNBC
NCT03330847	II	Recruiting	AZD6738, AZD1775	Olaparib		2nd/3rd line TNBC
NCT03428607	II	Active, not recruiting	AZD6738	Olaparib		Relapsed SCLC
NCT03462342	II	Recruiting	AZD6738	Olaparib		Recurrent ovarian cancer, platinum- sensitive or -resistant
NCT03682289	II	Recruiting	AZD6738	Olaparib		Metastatic renal cell, urothelial, pancreatic
NCT03787680	II	Recruiting	AZD6738	Olaparib		Metastatic castration-resistant prostate
NCT03878095	II	Recruiting	AZD6738	Olaparib		*IDH*-mutant solid tumors
NCT04065269	II	Recruiting	AZD6738	Olaparib		Relapsed *ARID1A*(-) or (+) gynecological cancers
NCT04239014	II	Not yet recruiting	AZD6738	Olaparib		Platinum-sensitive relapsed epithelial ovarian with previous PARPi
NCT04298021	II	Not yet recruiting	AZD6738	Olaparib	Durvalumab	Advanced cholangiocarcinoma
NCT04417062	II	Not yet recruiting	AZD6738	Olaparib		Recurrent osteosarcoma
*CHK1 inhibitor*						
NCT03057145	I	Active, notrecruiting	Prexasertib	Olaparib		Advanced solid tumors
*WEE1 inhibitor*						
NCT02511795	Ib	Completed	AZD1775	Olaparib		Refractory solid tumors
NCT02576444	II	Not yet recruiting	AZD1775	Olaparib		Advanced solid tumors with *p53/KRAS* mutation
NCT03579316	II	Recruiting	AZD1775	Olaparib		Recurrent ovarian, peritoneal, fallopian tube
NCT04197713	I	Not yet recruiting	AZD1775	Olaparib		Advanced solid tumors with previous PARPi

**Table 3 cancers-14-02640-t003:** Clinical trials evaluating TKI in combination with PARPi.

ClinicalTrials.gov Identifier [86] (accessed 20 May 2022)	Phase	Status	TKI	PARPi	Other Drugs	Cancer
*Pan-TKI*						
NCT01116648	I/II	Recruiting	Cabozantinib	Niraparib		Advanced urothelial
EGFR inhibitor						
NCT03891615	I	Recruiting	Osimertinib	Niraparib		*EGFR*-mutant advanced lung
VEGFR inhibitor						
NCT01116648	I/II	Active, not recruiting	Cediranib	Olaparib		Recurrent ovarian, fallopian tube, peritoneal, or triple negative breast cancer
NCT02340611	II	Completed	Cediranib	Olaparib		Recurrent ovarian with prior PARPi response
NCT02345265	II	Active, not recruiting	Cediranib	Olaparib		Recurrent ovarian, fallopian tube, or peritoneal
NCT02484404	I/II	Recruiting	Cediranib	Olaparib	Durvalumab	Advanced solid tumors
NCT02498613	II	Recruiting	Cediranib	Olaparib		Advanced solid tumors
NCT02502266	II/III	Recruiting	Cediranib	Olaparib		Recurrent platinum-resistant ovarian, fallopian tube, or peritoneal
NCT02681237	II	Active, not recruiting	Cediranib	Olaparib		Recurrent ovarian with prior PARPi response
NCT02893917	II	Active, not recruiting	Cediranib	Olaparib		Metastatic castration-resistant prostate
NCT02899728	II	Terminated	Cediranib	Olaparib	Platinum,	Extensive stage small cell lung
					etoposide	
NCT02974621	II	Recruiting	Cediranib	Olaparib		Recurrent glioblastoma
NCT03278717	III	Recruiting	Cediranib	Olaparib		Recurrent ovarian with prior platinum response
NCT03660826	II	Suspended	Cediranib	Olaparib		Metastatic endometrial
MEK inhibitor						
NCT03162627	I/II	Recruiting	Selumetinib	Olaparib		Advanced solid tumors
PI3K pathway inhibitors						
NCT02208375	Ib/II	Active, not recruiting	AZD5363 (AKT) or AZD2014 (mTOR)	Olaparib		Recurrent endometrial, ovarian, peritoneal, fallopian tube, or TNBC
NCT02511795	Ib	Completed	AZD1775 (PI3K)	Olaparib		Refractory solid tumors
NCT02576444	II	Active, not recruiting	AZD5363 (AKT)	Olaparib		Advanced solid tumors with *PTEN/PI3KCA/AKT/ARID1A* mutation
NCT03579316	II	Recruiting	AZD1775 (PI3K)	Olaparib		Recurrent ovarian, peritoneal, or fallopian tube
NCT04197713	I	Active, not recruiting	AZD1775 (PI3K)	Olaparib		Advanced solid tumors with prior PARPi response

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
