# Peer review of "Pharmacologic Induction of BRCAness in *BRCA*-Proficient Cancers: Expanding PARP Inhibitor Use"

_cancers, 2022, doi:10.3390/cancers14112640_

Round 1
Reviewer 1 Report
The manuscript entitled “Pharmacologic induction of BRCAness in BRCA-proficient cancers:
expanding PARP inhibitor use” is a very well prepared review on the therapeutic potential of PARP inhibitors. The authors have rationally structured this review and the cited literature is of good quality and quantity. It is also well written and it is easy to follow.
I have only two suggestions to the authors:
- I think that the mentions to clinical trials throughout the manuscrip lack proper citation. In only one case (line 353) the ClinicalTrials.gov. is mentioned (should be also included in the reference list) and only one trial number is used (line 508). I suggest to include a Table with all the clinical trials that the manuscript implies in order the reader to have a clearer picture of the number of relevant ongoing trials for each therapeutic scheme.
- I suggest to transfer section 4.7 as new 5.1 and correct the numbering of section 5 accordingly (Note that in the current version 5.1 (line 511) should be 5.2).
The authors should carefully re-examine the text and edit some errors (like in lines 315 (inhibition), 425 (an), 575 (considered)) and re-examine the acronyms. For instance I would prefer the use of PARPi instead of PARPis. In any case a uniform format should be used throughout the manuscript. Also some acronyms may not be explained (e.g. SSBR) or used (e.g. GIST).
Overall is an interesting manuscript with sufficient detail in presenting the interesting ongoing investigation of the putative combinatorial treatments leading to BRACAness and PARPi synthetic lethality.
Author Response
Response to Reviewer 1 Comments
Point 1: I think that the mentions to clinical trials throughout the manuscrip lack proper citation. In only one case (line 353) the ClinicalTrials.gov. is mentioned (should be also included in the reference list) and only one trial number is used (line 508). I suggest to include a Table with all the clinical trials that the manuscript implies in order the reader to have a clearer picture of the number of relevant ongoing trials for each therapeutic scheme.
Response 1: Tables have now been included to summarize the current relevant clinical trials.
Point 2: I suggest to transfer section 4.7 as new 5.1 and correct the numbering of section 5 accordingly (Note that in the current version 5.1 (line 511) should be 5.2).
Response 2: Suggestion has been followed and numbering updated/corrected.
Point 3: The authors should carefully re-examine the text and edit some errors (like in lines 315 (inhibition), 425 (an), 575 (considered)) and re-examine the acronyms. For instance I would prefer the use of PARPi instead of PARPis. In any case a uniform format should be used throughout the manuscript. Also some acronyms may not be explained (e.g. SSBR) or used (e.g. GIST).
Response 3: Text has been updated to correct the indicated (and other) minor errors.
Reviewer 2 Report
PARP inhibitors represent the novel approach to cancer treatment, driven by individualized decision-making. Initially, PARPi were used in patients with BRCA1/2 pathogenic alterations, advocating for a tumor specificity role once only tumor cells would be BRCA null. Over time, our comprehension of synthetic lethality grew, which led to the extension of its use – now embracing tumors with HR alterations. Which led us to this next stage, the use of PARPi combined with other drugs, mimicking an HR deficient phenotype.
This review summarized very well all combinations that may be used in clinical, and many of them are in clinical trials. Overall, this review presents the main PARPi inhibitors to induce a synthetic lethality phenotype.
Nevertheless, some points should be addressed.
Authors state (line 193) that "HDACis acetylate PARP1 directly", but reference [65] does not support that statement. According to Robert and colleagues, pan-HDACi (TSA) inhibits the deacetylation of PARP1. Therefore, authors should review this sentence and clarify the information regarding PARP1 acetylation.
Minor issues:
- Authors should clarify whether all FDA-approved PARPi are listed in the paper, and also whether other PARPi are in development/trial. Different inhibitors have different mechanisms of action, this also needs to be clarified.
- The authors should run a careful review of acronyms, some are not explained the first time they appear (e.g. DSB, first appearance in line 62, but it is only explained in line 139). Human proteins must be represented in capital letters (i.e., CHK, line 265; CDK1, line 278)
- The acronym for ATR is not "ATM and Rad3-related" (line 264) but ataxia-telangiectasia and Rad3-related.
- There are phrases and paragraphs without any reference. They are listed below:
- Line 78: "This powerful concept allows […] with few off target complications."
- The first paragraph of topic 4.3, lines 437 to 450.
- Line 467 – 471: "Vascular endothelial growth […] anti-angiogenic therapies."
- Line 512 – 518: "The phosphoinositide 3-kinase pathway […] proliferation phenotype."
- Line 533 – 535: "c-ABL tyrosine kinase […] 'Philadelphia chromosome'."
- According to the authors, "EZH2 is the histone methyltransferase subunit of PRC2, which methylates histones H3 on lysine 27", which agrees with the literature. However, in figure 1, the authors indicate EZH2 in a parallel with HDAC, which acts in acetylation. Could the authors clarify this point? Is there a role for EZH2 in acetylation?
- The authors should carefully review the figure's subtitles. The subtitles in the current version are poorly explored. They should be improved to guide the readers through the different pathways presented.
- All running trials are cited throughout the text, but the authors should indicate the name of each trial for reference.
Author Response
Response to Reviewer 2 Comments
Point 1: Authors state (line 193) that "HDACis acetylate PARP1 directly", but reference [65] does not support that statement. According to Robert and colleagues, pan-HDACi (TSA) inhibits the deacetylation of PARP1. Therefore, authors should review this sentence and clarify the information regarding PARP1 acetylation.
Response 1: The reference has been updated to correct the citation to Robert et al, and the text has been altered for clarity to “Notably, inhibition of deacetylation activity following HDACi exposure leads to PARP1 hyperacetylation and enrichment in chromatin that resembles PARPi-induced PARP trapping. When combined with PARPi, HDACi treatment further increases PARP trapping, synergistically sensitizing to the PARP-trapping PARPi talazoparib”.
Point 2: Authors should clarify whether all FDA-approved PARPi are listed in the paper, and also whether other PARPi are in development/trial. Different inhibitors have different mechanisms of action, this also needs to be clarified.
Response 2: FDA approvals have been edited for clarity, and PARPi in late stage development have been included. The reader is directed to Murai et al for precise discussion of catalytic inhibition vs. PARP trapping.
Point 3: The authors should run a careful review of acronyms, some are not explained the first time they appear (e.g. DSB, first appearance in line 62, but it is only explained in line 139). Human proteins must be represented in capital letters (i.e., CHK, line 265; CDK1, line 278)
The acronym for ATR is not "ATM and Rad3-related" (line 264) but ataxia-telangiectasia and Rad3-related.
Response 3: The suggested edits have been addressed.
Point 4: There are phrases and paragraphs without any reference. They are listed below:
- Line 78: "This powerful concept allows […] with few off target complications."
- The first paragraph of topic 4.3, lines 437 to 450.
- Line 467 – 471: "Vascular endothelial growth […] anti-angiogenic therapies."
- Line 512 – 518: "The phosphoinositide 3-kinase pathway […] proliferation phenotype."
- Line 533 – 535: "c-ABL tyrosine kinase […] 'Philadelphia chromosome'."
Response 4: Additional citations have been included at the indicated phrases.
Point 5: According to the authors, "EZH2 is the histone methyltransferase subunit of PRC2, which methylates histones H3 on lysine 27", which agrees with the literature. However, in figure 1, the authors indicate EZH2 in a parallel with HDAC, which acts in acetylation. Could the authors clarify this point? Is there a role for EZH2 in acetylation?
Response 5: EZH2 inhibitor should indicate a role in methylation, not acetylation. The figure has been updated to correct this error.
Point 6: The authors should carefully review the figure's subtitles. The subtitles in the current version are poorly explored. They should be improved to guide the readers through the different pathways presented.
Response 6: Figure legends have been expanded for improved clarity.
Point 7: All running trials are cited throughout the text, but the authors should indicate the name of each trial for reference.
Response 7: Tables have now been included to summarize the current relevant clinical trials.
Reviewer 3 Report
Dear Authors,
I’m writing here to submit a revision for an article that I have recently received. The article “Pharmacologic Induction of BRCAness in BRCA-Proficient Cancers: Expanding PARP Inhibitor Use” by R. Abbotts et al., aimed on highlitening of past and the most recent finding in the mechanism of action of anticancer agents known as PARP inhibitors.
I have only one important suggestion regarding the Introduction part.
- I would suggest rearranging the information in the Introduction. It could be more logical to start with by explaining that the several groups of anti-cancer therapy are based on genotoxic effects based on induction DSB and here introduce the term "BRCAness" as class of cancers that have defects in double-strand break repair (DSBR) by homologous recombination repair (HRR). Then turn to PARP inhibitors in general and introduce synthetic lethality phenomenon in BRCA-deficient cancers, and why it makes PARPi as drug of choice for "BRCAness" tumors. After that focus on PARP itself as an enzyme playing an essential role in repair of SSB. And discussion at the end of introduction about PARP and PARP related PTMs itself could serve as connection to other pathways that are dependent on PARP activity/ potential PARPi targets.
Here is some minor comments:
- I would suggest removing the term BRCAness from the Simple Summary and Abstract, and replace cancer with "defects in double-strand break repair (DSBR) by homologous recombination repair (HRR)".
2. Line 11. BRCA to BRCA . And gene names in general in other sentences.
3. change "Tyrosine kinase signaling pathways (Figure 3)"to "Tyrosine kinase signaling pathways"
Author Response
Response to Reviewer 3 Comments
Point 1: I would suggest rearranging the information in the Introduction. It could be more logical to start with by explaining that the several groups of anti-cancer therapy are based on genotoxic effects based on induction DSB and here introduce the term "BRCAness" as class of cancers that have defects in double-strand break repair (DSBR) by homologous recombination repair (HRR). Then turn to PARP inhibitors in general and introduce synthetic lethality phenomenon in BRCA-deficient cancers, and why it makes PARPi as drug of choice for "BRCAness" tumors. After that focus on PARP itself as an enzyme playing an essential role in repair of SSB. And discussion at the end of introduction about PARP and PARP related PTMs itself could serve as connection to other pathways that are dependent on PARP activity/ potential PARPi targets.
Response 1: We thank the reviewer for this suggestion. While we see the merit in the suggested edit, we have elected not to modify the structure of the introduction at this time. In composing our original submission, we wished to reflect the timeline of PARP inhibitor development, from protein target identification, to inhibitor development, to subsequent discovery of synthetic lethality, to therapeutic expansion into the BRCAness domain. We believe that by following historical PARPi development, the current structure provides an equally logical introduction to the topic.
Point 2: I would suggest removing the term BRCAness from the Simple Summary and Abstract, and replace cancer with "defects in double-strand break repair (DSBR) by homologous recombination repair (HRR)".
Response 2: We have elected to retain references to BRCAness in the simple summary and abstract, given that the title of the review includes this term. However, the simple summary has been edited to improve clarity, including edits in line with the suggestion.
Point 3: Line 11. BRCA to BRCA . And gene names in general in other sentences.
Response 3: Gene names have been italicized as required.
Point 4: change "Tyrosine kinase signaling pathways (Figure 3)"to "Tyrosine kinase signaling pathways
Response 4: This suggestion has been accepted and the text modified accordingly.